# Alpha-Synuclein Contribution to Neuronal and Glial Damage in Parkinson’s Disease

**DOI:** 10.3390/ijms25010360

**Published:** 2023-12-26

**Authors:** Kamil Saramowicz, Natalia Siwecka, Grzegorz Galita, Aleksandra Kucharska-Lusina, Wioletta Rozpędek-Kamińska, Ireneusz Majsterek

**Affiliations:** Department of Clinical Chemistry and Biochemistry, Medical University of Lodz, 92-215 Lodz, Poland; kamil.saramowicz@stud.umed.lodz.pl (K.S.); natalia.siwecka@stud.umed.lodz.pl (N.S.); grzegorz.galita@umed.lodz.pl (G.G.); ola_kucharska@wp.pl (A.K.-L.); wioletta.rozpedek@umed.lodz.pl (W.R.-K.)

**Keywords:** Parkinson’s disease, α-synuclein, neurodegeneration, Lewy bodies, aggregation, neurons, glial cells, neuroinflammation, seeding

## Abstract

Parkinson’s disease (PD) is a complex neurodegenerative disease characterized by the progressive loss of dopaminergic neurons in the substantia nigra and the widespread accumulation of alpha-synuclein (αSyn) protein aggregates. αSyn aggregation disrupts critical cellular processes, including synaptic function, mitochondrial integrity, and proteostasis, which culminate in neuronal cell death. Importantly, αSyn pathology extends beyond neurons—it also encompasses spreading throughout the neuronal environment and internalization by microglia and astrocytes. Once internalized, glia can act as neuroprotective scavengers, which limit the spread of αSyn. However, they can also become reactive, thereby contributing to neuroinflammation and the progression of PD. Recent advances in αSyn research have enabled the molecular diagnosis of PD and accelerated the development of targeted therapies. Nevertheless, despite more than two decades of research, the cellular function, aggregation mechanisms, and induction of cellular damage by αSyn remain incompletely understood. Unraveling the interplay between αSyn, neurons, and glia may provide insights into disease initiation and progression, which may bring us closer to exploring new effective therapeutic strategies. Herein, we provide an overview of recent studies emphasizing the multifaceted nature of αSyn and its impact on both neuron and glial cell damage.

## 1. Introduction

Parkinson’s disease (PD) is the second-most common age-related neurodegenerative disease after Alzheimer’s. It represents the fastest-growing neurological condition, affecting over 8.5 million individuals worldwide [1]. PD is clinically characterized by primary motor features as well as secondary non-motor features. Motor features of PD include resting tremor, bradykinesia, muscle rigidity, and postural instability. On the other hand, non-motor features comprise hyposmia, autonomic dysfunction (e.g., orthostatic hypotension, constipation, erectile dysfunction, and sialorrhea), sleep disorders, sensory impairment, neuropsychiatric symptoms (e.g., mood disturbances, depression, and psychotic episodes), and cognitive impairment [2]. Non-motor symptoms may occur prior to motor features, and therefore they can be considered a prodromal state of the disease, with occurrence even up to 20 years before clinical diagnosis of PD [3].

PD is characterized by the progressive loss of dopaminergic (DA) neurons in the substantia nigra pars compacta (SNpc) of the midbrain with the concurrent decline in nigrostriatal DA neurotransmission. A key pathological hallmark of PD is the presence of intracellular protein inclusions known as Lewy bodies (LBs) and Lewy neurites (LNs), the main component of which is misfolded and aggregated protein alpha-synuclein (αSyn). Abnormal accumulation and spreading of toxic αSyn aggregates are the major molecular events underlying PD pathogenesis [4]. Therefore, PD has been classified as a heterogenous group of diseases called α-synucleinopathies. PD is the most prevalent among α-synucleinopathies, while other known forms include dementia with Lewy bodies (DLB), PD with dementia (PDD), multiple system atrophy (MSA), and neuroaxonal dystrophy [5]. Differences in affected brain regions, cell types, and genetic factors reflect the diverse manifestations observed in different α-synucleinopathies. The heterogeneous nature of α-synucleinopathies may also be attributed to the distinct characteristics of different conformational αSyn strains found in neurons and glia [6]. These properties of αSyn strains may serve as a foundation for distinguishing between various α-synucleinopathies, such as PD and MSA [7]. The αSyn aggregates contribute to various cellular dysfunctions, including impaired mitochondrial function, ER stress, disruption in the autophagy–lysosomal pathway (ALP), as well as synaptic and nuclear dysregulation. DA neurons of the SNpc are particularly vulnerable to αSyn-induced toxicity [8]. However, studies suggest that the onset of pathological aggregation potentially begins outside the CNS. The pathological process is believed to start in the olfactory bulb or enteric nervous system (ENS) and then spread retrogradely to the amygdala or brainstem [9], which may give an explanation for some prodromal features of PD. Moreover, neuronal disturbances and LBs can be found in other brain regions such as the locus coeruleus, pedunculopontine nucleus, Meynert’s nucleus, and raphe nucleus [10,11], contributing to the impairment of noradrenergic, cholinergic, and serotonergic pathways. Recent studies suggest that αSyn accumulation extends beyond the brain, as it manifests in various organs outside the CNS. Studies have identified αSyn deposits in the gastrointestinal tract, liver, salivary glands, and skin. αSyn has also been detected in body fluids such as blood, cerebrospinal fluid (CSF), saliva, and tears. These findings not only reinforced the view that PD is a systemic disorder but also pointed to the possibility of using peripheral, readily obtainable αSyn as a potential biomarker of PD [12,13,14]. Emerging evidence suggests that the ubiquity of αSyn may be attributed to its capacity to propagate from cell to cell in a prion-like manner. This hypothesis proposes that misfolded αSyn seeds are secreted from donor cells to the extracellular matrix (ECM) and then taken up by adjacent cells [15]. The seeding capacity of endogenous αSyn is contingent upon the origin of neurons and their levels of αSyn expression [16]. However, mounting evidence has demonstrated that glial cells may play a significant role in the progression of αSyn pathology. While astrocytes and microglia can contribute to neuroinflammation and neuronal damage, they may also be involved in neuroprotection and clearance of pathological αSyn [17,18,19]. Thus, astrocytes and microglia play a dual role in PD, acting as a double-edged sword.

The genetic background of PD also underscores the key role of αSyn in neurodegeneration, as most of the genes implicated in PD pathology are related to αSyn synthesis, trafficking, and clearance. The *SNCA* gene is located on chromosome 4q21.3-22 and encompasses five exons that encode for αSyn [20]. It was the first causative gene identified in the monogenic form of PD [21]. Point mutations in the N-terminal region (e.g., A30P, E46K, H50Q, G51D, A53E, and A53T) are strongly associated with autosomal dominant PD, while duplications and triplications of the *SNCA* gene are associated with familial PD cases characterized by early onset and rapid progression [22]. Additionally, single nucleotide polymorphisms in *SNCA* have been identified as a significant risk factor for sporadic PD [23]. Studies of familial PD have identified other causative genes implicated in the autosomal dominant form of the disease (e.g., *LRRK2*—encoding leucine-rich repeat kinase 2 and *VPS35*—encoding vacuolar protein sorting 35) or autosomal recessive (e.g., *PRKN*—encoding Parkin RBR E3 Ubiquitin Protein Ligase, *PINK1*—encoding PTEN-induced kinase 1, *ATP13A2*—encoding lysosomal type 5 P-type ATPase, *PLA2G6*—encoding phospholipase A2, and *FBX07*—encoding F-box protein 7) [24]. In addition, there are some genetic risk factors that predispose to PD, such as *GBA1* mutations, which reduce the activity of glucocerebrosidase (a lysosomal enzyme encoded by this gene). As a result, lysosomal degradation of αSyn is significantly impaired [25]. Mutations in the aforementioned genes apparently enhance αSyn-induced cellular damage, and this will be discussed in the forthcoming sections.

To date, there is no neuroprotective or neurorestorative therapy for the treatment of PD. Despite advances in unraveling the pathogenesis of PD, the exact mechanisms underlying the progressive loss of DA cells in SNpc remain elusive. Considering the importance of αSyn in the development and progression of PD, this review summarizes current knowledge on the nature of various αSyn species and their involvement in neurotoxicity. We present evidence for multiple avenues leading to αSyn-induced cellular dysfunction, such as impaired mitochondrial function, loss of protein homeostasis, and nuclear dysfunction. Finally, we discuss the current concept regarding αSyn propagation and the role of glia and neuroinflammation in the course of PD progression.

## 2. The Structure and Role of αSyn

### 2.1. αSynuclein Structure

αSyn is an intracellular, 14kDa protein composed of 140 amino acids. The primary αSyn amino acid sequence comprises three regions: N-terminal, NAC (non-amyloid β-component), and C-terminal, each exhibiting distinct molecular and biological properties [26]. In healthy neurons and other types of cells, αSyn primarily exists in the form of unfolded monomers [27,28,29]. Monomeric αSyn is considered an intrinsically disordered protein due to its lack of native three-dimensional structure. Therefore, it is characterized by structural flexibility and susceptibility to modifications, which makes it prone to misfolding and aggregation [30]. Due to its flexibility, monomeric αSyn is able to dynamically interconvert between various conformational states (e.g., α-helix, β-sheet, and random-coil), which is influenced by environmental conditions and the molecular characteristics of αSyn regions [31,32].

The amphipathic N-terminal region (residues 1–60) tends to adopt an α-helix structure and acts as an anchor for lipid membrane binding [33]. The essential structural element of these interactions constitutes seven imperfect 11-residue repeat sequences, which comprise the core consensus motif “KTKEGV”. The N-terminus contains four 11-residue repeat sequences, while the remaining three are located in the NAC region. The lysine residues within this motif interact with the negatively charged phospholipid head groups of the lipid membrane [34]. Subsequently, αSyn forms α-helix on lipid membranes, which influences the composition and curvature of lipid bilayers and participates in the regulation of synaptic vesicle clustering and docking [35].

The centrally located hydrophobic NAC region (residues 61–95) is involved in the regulation of αSyn transport within axons [36] and exhibits a propensity for adopting a cross-β-sheet structure. This characteristic promotes the aggregation process by recruiting additional monomers to form higher molecular weight assemblies, which eventually leads to the formation of amyloid fibrils [37]. Especially, the amino acid residues located in the core of NAC play a crucial role in the pathological assembly of αSyn into toxic filaments. Deletion of the 12-amino acid sequence motif “VTGVTAVAQKTV” (residues 71–82) results in the complete abolition of amyloid formation [38]. Also, NACore (residues 68–78) or the SubNACore (residues 69–77) regions have been implicated in the formation of highly organized parallel β-sheet structures, leading to their spontaneous self-assembly and exertion of cytotoxic effects [39].

Although the NAC region alone is recognized as necessary and sufficient for αSyn fibrillization [38,39], recent studies have shown that both the N-terminal and C-terminal regions are strictly involved in the kinetic regulation of this process. Specific sequence motifs embedded in the distal part of the N-terminal region, P1 (residues 36–42) and P2 (residues 45–57), modulate the conformational properties of monomeric αSyn that favor aggregation [40]. Upon calcium binding to the C-terminus, conformational transitions are initiated, increasing the exposure of the N-terminal region, which increases the susceptibility of the αSyn monomers to self-assembly [41].

The C-terminal region (residues 95–140) is involved in calcium-mediated synaptic vesicle binding, thereby promoting membrane fusion and neurotransmitter release [42,43]. Structurally, it displays a flexible random coil arrangement [44]. It is enriched in acidic amino acids, exhibiting high negative charge density and increased polarity [45]. The C-terminus engages in long-range electrostatic interactions with both the N-terminus and NAC region, which stabilize the monomeric αSyn structure [46,47,48]. While interacting with the N-terminus, the negatively charged C-terminus provides an electrostatic shield that prevents NAC from engaging in cross-β-sheet folding. Truncation of the C-terminus may reflect decreased charge repulsion and increased exposure of the NAC region, thus promoting a pro-aggregatory conformational state [49].

As monomeric αSyn is intrinsically disordered and susceptible to undergoing pathological aggregation, a plethora of studies have investigated potential mechanisms underlying the maintenance of its structural stability. It has been observed that upon binding to the lipid membranes and adopting α-helical structure, αSyn readily multimerizes into high-order species to execute its intrasynaptic functions [50]. In addition to the interaction of αSyn with lipids, studies have shown that in solution, αSyn monomers combine to form stable tetramers that resist aggregation. In contrast to monomeric αSyn, tetrameric forms exhibit spontaneous random coil to α-helical conversion, presenting a higher degree of native α-helicity [51,52]. Deletion of the “KTKEGV” motif abolishes tetramerization, which highlights the role of this motif in consolidating the α-helix structure in the absence of lipids [53]. The physiological significance of whether the monomeric or tetrameric forms predominate remains a subject of extensive debate. However, it is believed that both forms coexist in a dynamic equilibrium, and an imbalance in the tetramer:monomer ratio can affect the potential for aggregation [30]. Various Parkinson-causing mutations, including point mutations of the *SNCA* (e.g., A53T and E46K), *LRRK2* (e.g., G2019S and R1441C), and *GBA1* (e.g., N370S and L444P), induce destabilization of the αSyn tetrameric structure; this leads to an increase in monomeric forms, potentiates inclusion formation, and induces neurotoxicity [54,55,56]. Conversely, maintenance of proper lysosomal integrity (adequate GBA1 expression) and lipid homeostasis (increased saturated and decreased unsaturated fatty acid levels) stabilizes tetramers and protects cells against αSyn-induced toxicity [57,58,59].

Furthermore, studies have discovered that under physiological conditions, αSyn monomers are N-terminally acetylated, which stably preserves their structure and prevents spontaneous aggregation [60]. N-terminal acetylation of monomeric αSyn increases its interaction with lipid vesicle membranes, thereby delaying aggregation [61]. Therefore, acetylation of the N-terminal can be considered a protective modification against adopting an amyloid-like β-sheet structure by αSyn as well as the proliferation of pathogenic aggregates [62].

### 2.2. αSyn Aggregation and Variety of αSyn Species

The common paradigm among proteinopathies suggests that the initiation of pathological aggregation takes place when a native protein transforms into a pathogenic conformation. This conformation then prompts an endogenous protein to adopt the same structure, initiating self-assembly [63]. To date, the direct mechanism of αSyn aggregation remains unclear. However, it is believed that αSyn enters a multi-step process in which an unfolded monomer undergoes primary nucleation, elongation, and secondary nucleation. During primary nucleation, the self-assembly of monomers contributes to the formation of soluble oligomeric intermediates (nuclei), which may occur both in the solution and on the lipid surface [64,65]. Nuclei are defined as the smallest aggregates, which are kinetically stable enough to promote further growth by the addition of monomers more readily than dissociation. Elongation is the process of adding monomers to the ends of existing aggregates, starting from the nuclei and leading to the formation of highly organized structures [66]. It has also been demonstrated that the self-association of oligomers can seed further aggregation, and this appears to be a more effective pathway of elongation than monomer addition [67]. The products of elongation, such as protofibrils and fibrils, can further constitute nucleation centers and provide autocatalytic amyloid amplification. This process is called secondary nucleation, and due to the interaction of the monomeric domain with the fibril surface, it contributes to the rapid generation of new aggregates [68]. The initial dimerization of αSyn at the membrane surface is proposed as the primary driver of oligomerization [69,70]. This process begins with the formation of α-helical-rich intermediate species [71]. As the transformation into fibrils proceeds, the α-helical content diminishes in favor of the β-sheet structure. β-sheet-like interactions stabilize the oligomeric structure and promote the buildup of higher-molecular-weight insoluble fibrils. In contrast to mature fibrils, which exhibit a parallel β-sheet structure, prefibrillar oligomers (protofibrils) predominantly feature an anti-parallel β-sheet configuration, which potentiates their toxic properties [72]. There is a growing consensus in the field that small αSyn oligomers and their distinct conformations are likely toxic, disease-causing species [73]. However, not all oligomers exhibit seeding competence, indicating their very limited ability to participate in the elongation process. These species consisting of 15–30 protein molecules and adopting a spherical appearance are called off-pathway oligomers [74,75,76]. These oligomers are kinetically trapped, which means they are thermodynamically unstable but persist due to kinetic barriers that prevent further progression or reversion to a more stable state. These trapped states represent local energy minima that temporarily halt the aggregation process, resulting in the accumulation of partially folded or misfolded protein species [77]. Once oligomers enter the cells, they disrupt membranes, generate reactive oxygen species (ROS), elevate cytosolic calcium levels, and trigger cell death [74,78]. On the other hand, the on-pathway oligomers are seeding-competent and have a wreath-like appearance similar to other amyloid protofilaments [79]. Recently, both on-pathway and off-pathway oligomers have been found to be derived from a common metastable precursor, a spherical oligomer composed of 11 monomers [80]. Albeit small αSyn oligomers are considered critical species driving PD progression (as described in depth in [81]), their direct study poses many difficulties given their transient, metastable characteristics and high level of heterogeneity. In contrast, recent advances in high-resolution techniques have allowed for an in-depth exploration of fibril structures. For example, cryogenic electron microscopy studies have revealed distinct αSyn fibril polymorphs [82]. The predominant polymorphs observed are “rods” and “twisters”, both formed by pairs of β-arch protofilaments that share a common structural kernel of a bent β-arch. These polymorphs primarily differ in packing arrangement and the structure of each αSyn molecule. In the twister structure, each molecule forms a bent β-arc with an NACore interface (residues 68–78). In contrast, the rod structure incorporates additional ordered residues, leading to the formation of a “Greek key” supersecondary structure [82,83,84,85]. Importantly, different strains of αSyn with distinct morphologies exhibit diverse seeding capacities and lead to specific pathological features and neurotoxic effects that are unique to each strain [86,87]. For example, strains found in MSA exhibit a higher proportion of β-sheet, which reflects their higher seeding activity in comparison with strains observed in PD [6,7]. These observations have prompted the development of new methods for differentiating α-synucleinopathies, the overlapping features of which have often posed a major diagnostic challenge. According to the newest reports, the seed amplification assay (SAA) may serve as a highly sensitive and specific tool for identifying individuals with PD, even in the prodromal stages of the disease, while also providing insights into the molecular diversity of the condition [88]. The method takes advantage of the self-replicating nature of small αSyn nuclei, allowing them to elongate in vitro into easily detectable fibrils [89]. These approaches offer opportunities for highly accurate molecular diagnosis of α-synucleinopathies, shedding light on the importance of how biophysical research translates into precision medicine achievements.

### 2.3. Mechanisms Implicated in αSyn Modifications

The actual factors that initiate αSyn aggregation are largely unknown. However, the intrinsic structural properties of the protein, its intramolecular interactions, and external physical and biochemical factors are thought to kinetically regulate the aggregation process. Missense point mutations in the SNCA gene alter the structural properties of αSyn, which increases its potential for misfolding and aggregation but also affects the conformation of αSyn fibrils [90]. Likewise, several external factors increase the propensity of αSyn to aggregate. These factors include molecular crowding, acidic pH, temperature, the presence of metal ions, lipids, toxins, and ROS [91]. Moreover, the interactions of αSyn with its microenvironment are influenced by post-translational modifications (PTMs), which have been established as key determinants of αSyn pathology [92]. Indeed, studies from human postmortem samples have consistently shown that phosphorylated and ubiquitinated forms of αSyn are common pathological features in LBs and LNs [93,94]. Other PTMs, such as nitration and C-terminal truncation, have also been identified within the neuronal and glial inclusions of α-synucleinopathies patients [95,96,97,98]. Moreover, PTMs like SUMOylation and O-GlcNAcylation have been investigated in animal and cell culture models of PD [98,99]. All these modifications can either promote or impede αSyn aggregation, depending on the specific residue that is modified. Nevertheless, O-GlcNAcylation, similar to acetylation, has been frequently associated with reduced αSyn aggregation and attenuated PD-related toxicity [99,100]. Certain PTMs and species of αSyn are detectable in CSF and plasma, so they could serve as potential biomarkers for PD and other α-synucleinopathies [92]. Yet, it is unclear to what extent levels and modifications of αSyn in peripheral fluids reflect its status in the CNS or relate to disease progression or severity. Herein, we present a concise outline of the most relevant mechanisms involved in αSyn modifications.

#### 2.3.1. Mutations

Mutations of αSyn are utterly restricted to the N-terminal domain (e.g., A30P, A53T, E46K, G51D, A53E, and H50Q). The A30P mutation promotes oligomerization but not fibrillation compared to the wild-type protein (WT) [101]. A53T has the greatest increase in the αSyn oligomer accumulation, but it has no impact on the formation of the insoluble fibrillar inclusions. On the contrary, A30P, E46K, and G51D drastically increase the level of inclusions while lowering the oligomer rate [102]. Although the A53E mutation hinders aggregate formation compared to WT, it alters fibril morphology, leading to increased toxicity [103]. Similarly, H50Q polymorphs exhibit greater cytotoxicity, along with higher aggregation rates and seeding capacity [104].

#### 2.3.2. Ubiquitination

Ubiquitination is a posttranslational modification crucial for protein degradation via the ubiquitin-proteasome system (UPS; mainly soluble αSyn) or macro-autophagy (αSyn oligomers and aggregates) [105]. αSyn ubiquitination primarily occurs within the N-terminus, and its effect on aggregation is dependent on the binding residue. Ubiquitination at Lys10 and Lys23 residues (N-terminus) promotes the formation of inclusions, whereas ubiquitin binding to other sites inhibits aggregation [106]. Interestingly, studies have shown that ubiquitination mediated by a specific type of E3-ubiquitin ligase SIAH-1 (seven in absentia monologue-1) does not target αSyn degradation by UPS but rather promotes its aggregation and toxicity [107].

#### 2.3.3. SUMOylation

SUMO (small ubiquitin-like modifier) is conjugated to αSyn at Lys residues, primarily within the C-terminus [108]. Similar to ubiquitination, SUMOylation seems to have divergent effects on αSyn distribution. PIAS2 (E3 SUMO-protein ligase)-mediated SUMOylation (by SUMO1) of αSyn promotes its aggregation directly or indirectly by blocking ubiquitin-dependent degradation. Both PIAS2 and SUMO1 were found to be elevated in the SN of PD brains [98]. Conversely, SUMOylation of αSyn at Lys96 or Lys102 residues inhibits aberrant aggregation [109]. These disparities may be attributed to the heterogeneous profile of SUMOylation and emphasize the need for further research.

#### 2.3.4. Phosphorylation

In contrast to acetylation, phosphorylation of αSyn is mainly restricted to the C-terminus. The most well-known αSyn phosphorylation occurs at the residue Ser129 (pS129-αSyn), and it is found abundantly within the LBs (up to 90% of inclusions) [93]. It has also been shown that phosphorylation of Ser129 increases the level of soluble oligomers, while phosphorylation of Tyr125 decreases its level [110]. Additionally, αSyn can be phosphorylated at Ser87 and Tyr136, both of which tend to have protective effects against αSyn aggregation [111,112].

#### 2.3.5. Nitration

It has been observed that nitration of the Tyr39 residue decreases the binding of αSyn monomers to lipid vesicles and reduces their ability to adopt α-helical conformation, thereby accelerating fibril formation [113]. Besides, nitration of αSyn at Tyr125 promotes the formation of αSyn dimers, which can further enhance the elongation process toward fibril formation [114].

#### 2.3.6. Truncation of the C-Terminal

This phenomenon has been estimated to occur in 10-30% of total αSyn within LBs, and it is considered highly neurotoxic as it enhances secondary nucleation processes [115]. αSyn truncated at residues 103 and 119 can act as a seed that promotes aggregation due to electrostatic effects; this facilitates autocatalytic secondary nucleation and proliferation of fibrils [116]. C-terminal truncated forms of αSyn, especially at residue 121, exhibit accelerated kinetics of aggregation and a higher cytotoxic effect compared to the full-length protein. Mechanistically, caspase-1 performs truncation of αSyn at the C-terminal to produce αSyn121 and induces neuroinflammation, which aggravates cytotoxicity and creates a positive-feedback loop [117]. Besides, recent studies have shown that truncation ranging from 121 to 125 residues increases fibrillation while retaining the ability to form toxic oligomers [49]. Overall, studies emphasize the importance of C-terminal truncation in the formation of abnormal αSyn species.

### 2.4. The Significance of Synaptic αSyn Interactions in Health and Disease

αSyn is an abundant presynaptic protein, preferentially anchored to the membrane of the synaptic vesicle (SV) due to the composition and physical properties of the membrane bilayer (e.g., curvature and charge) [117]. Despite extensive research, the precise function of αSyn is not fully understood. Yet, the preferential localization of αSyn and its strong affinity for synaptic proteins suggest a potential role in regulating important synaptic processes, such as synaptic vesicle trafficking, neurotransmitter release, and synaptic plasticity. Native αSyn potentially acts as a neuroprotective agent, reciprocally interacting with molecular chaperones (e.g., 14-3-3 proteins and cysteine-string protein-α), downregulating proapoptotic factors (e.g., protein kinase C δ and Bcl-2 associated death promoter), and supporting neurogenesis [118,119,120,121,122]. Consistently, loss-of-function of the native protein leads to synaptic disturbances, which have been proposed as one of the earliest molecular events driving PD [4].

Given that PD is defined by the deterioration of DA neurotransmission, many studies have indicated the role of αSyn in modulating DA signaling. αSyn-knockout mice exhibit reduced DA striatal levels and impaired DA-dependent motor responses [123]. Consistently, the absence of αSyn expression leads to selective loss of neurons in SNpc [124]. On the other hand, overexpression of αSyn directly impairs DA synthesis, reuptake, and release. In vitro studies have shown that αSyn overexpression leads to downregulation of crucial enzymes involved in DA biosynthesis (tyrosine hydroxylase and aromatic amino acid decarboxylase), as well as an impairment of dopamine transporter (DAT) reuptake activity [125,126,127]. Transgenic rodent models revealed that αSyn overexpression impairs synaptic vesicle clustering and membrane fusion, leading to a disruption in the exo/endocytic cycle. This, in turn, reduces DA release and contributes to rapid and extensive axonal damage that precedes neuronal death [128,129,130]. Although transgenic rodent models of αSyn overexpression are essential for PD research, they face limitations in the vague distinction between the buildup of physiological forms and pathological aggregates in the observed phenotypes [131]. Higher levels of αSyn expression potentiate aggregation in a dose-dependent manner, which leads to progressive neurodegeneration [132]. Yet, brain regions particularly susceptible to LB pathology, such as SNpc, do not display the highest endogenous expression of αSyn, which may suggest that αSyn levels alone do not exclusively determine cell vulnerability [133].

Mechanistically, αSyn acts as a molecular chaperone, interacting directly with VAMP2 (vesicle-associated membrane protein 2) and SNARE (soluble N-ethylmaleimide attachment protein receptor) complex assembly, which is essential to promote synaptic vesicle (SV) fusion at the presynaptic terminal. In nigral neurons, high-molecular-weight strains of αSyn were shown to interfere with SNARE complex formation by preferentially binding VAMP2. This impairs synaptic vesicle motility and reduces DA release [134]. αSyn and synapsin III cooperate in the regulation of synaptic arrangement and DA release via complex formation on the external membrane of SV. The absence of αSyn as well as its pathological aggregation led to an increased density of synapsin III in presynaptic boutons and reduced DA release [135]. Furthermore, under physiological conditions, αSyn interacts with vesicular monoamine transporter 2 (VMAT-2) to regulate DA storage in vesicles. αSyn upregulation inhibits VMAT-2 activity, which impairs DA sequestration and leads to neuronal damage from toxic DA metabolites and ROS [136]. In addition, the presence of mutant oligomers is associated with axonal transport dysfunction, decreased levels of synaptic proteins (e.g., synapsin I) and vesicles, and, as a consequence, dendrite and synapse loss [137,138]. These findings suggest that αSyn plays a multifaceted role in not only maintaining synaptic homeostasis but also emphasizing how loss-of-function of the native protein contributes to synaptic dysfunction (Figure 1).

## 3. αSyn-Induced Toxicity in Distinct Organelles

### 3.1. αSyn in the Mitochondria

Mitochondria are crucial organelles for ATP synthesis, Ca^2+^ ion storage, regulation of cellular metabolism and proliferation, as well as induction of apoptosis. Some studies suggest that, under physiological conditions, αSyn can positively regulate mitochondrial functioning. This positive effect is thought to be a result of several factors, including an increase in ATP synthase activity, prevention of lipid membrane peroxidation, preservation of mitochondrial fusion, and inhibition of mitochondria-mediated apoptosis [139,140,141,142]. However, this effect is only seen at moderate levels of αSyn monomers. Increased levels of αSyn can instead trigger mitochondrial dysfunction, particularly in the nigrostriatal DA neurons [143]. Conversely, impairment of mitochondrial functioning can aggravate αSyn accumulation, which initiates a vicious cycle implicated in PD pathogenesis [144]. DA neurons, with their high energy demands and chronic exposure to oxidative stress, are particularly vulnerable to mitochondrial damage. It was found that the SN contains twice as many oxidized proteins as compared to the striatum and frontal cortex in healthy individuals [145]. Accumulation of ROS further promotes the formation of αSyn oligomers, a reduction in ATP-synthase activity, as well as damage to mtDNA, which leads to alteration of mitochondrial gene expression [146,147,148]. ROS-induced mtDNA deletions contribute to diminished synthesis of ETC subunits, thereby intensifying ROS production [148]. SN is particularly susceptible to adopting mtDNA mutations, and their prevalence increases with aging. SN in healthy individuals displays a compensatory mechanism involving the upregulation of non-mutated mtDNA, which is abolished in PD patients [149].

αSyn binds to protein complexes located in the outer mitochondrial membrane (OMM), such as the translocase of the outer membrane (TOM; channel responsible for protein uptake from the cytosol) and the voltage-dependent anion channel (VDAC; regulator of Ca^2+^ homeostasis) [150,151]. TOM and VDAC can translocate monomeric αSyn into mitochondria. However, αSyn oligomers and pS129-αSyn block the TOM complex, thus inhibiting protein influx into mitochondria and consequently reducing electron transport chain (ETC) functioning [150]. αSyn monomers can also be imported into mitochondria via VDAC, yet their accumulation inhibits VDAC-mediated translocation [151]. Both TOM and VDAC elevate αSyn levels in the mitochondria, wherein they can inhibit complex I of ETC, which leads to the impairment of ATP synthesis and the accumulation of ROS (mainly H_2_O_2_) [152,153]. αSyn interaction with complex I is reciprocal, as inhibitors of complex I, including MPTP, rotenone, and paraquat, promote αSyn aggregation, and their administration causes permanent parkinsonian symptoms and DA cell loss [154,155].

Inside the mitochondrion, αSyn exhibits a high affinity for the major phospholipid constituent of mitochondrial membranes, cardiolipin, which is particularly abundant within the inner mitochondrial membrane (IMM) [156]. Cardiolipin initiates the oligomerization of A53T αSyn, which leads to an overproduction of mitochondrial ROS, further promoting αSyn oligomerization. Consequently, A53T αSyn oligomerization disrupts complex I function, hinders ATP production, and triggers the opening of the mitochondrial permeability transition pore (PTP) [157]. Oligomeric αSyn may induce PTP through numerous mechanisms (e.g., oligomer-induced oxidative stress or calcium dysregulation) [158]. PTP formation leads to impairment of ionic balance, mitochondrial swelling, membrane rupture, and cytochrome C leakage, which results in cell death [159].

Alterations in calcium homeostasis are another plausible event that contributes to mitochondrial failure and impairment of electrophysiological activity in DA neurons [160]. Mitochondrial Ca^2+^ uptake is primarily controlled by a network of calcium channels that remains reversibly tethered to a band of membrane proteins in the ER. These contact sites are known as mitochondria-associated ER membranes (MAMs) and, in addition to regulating calcium homeostasis, are involved in various cellular processes, including lipid biosynthesis, mitochondrial and ER dynamics, autophagy, and maintenance of cell survival [161]. As aberrant αSyn localizes in MAMs [162], several protein interactions involved in MAM formation are impaired. For example, vesicle-associated membrane protein B (VAPB) in the ER membrane forms a complex with protein tyrosine phosphatase-interacting protein-51 (PTPIP51) in the OMM. Overexpression of both WT and mutant αSyn in DA neurons reduces ER-mitochondria tethering through interaction with the VAPB–PTPIP51 complex, which leads to impaired Ca^2+^ exchange and ATP production [163]. Another important component of the MAM calcium homeostasis apparatus is the ER Ca^2+^-ATPase SERCA. Both soluble and insoluble αSyn aggregates, but not monomers, can bind and activate SERCA, resulting in decreased Ca^2+^ concentration in the cytosol, which ultimately leads to cell death [164]. Furthermore, genes linked with familial PD have also been associated with MAMs. The DJ-1 (Parkinson’s disease protein 7) mutation associated with familial PD was reported to reduce MAM association and disrupt calcium homeostasis, as DJ-1 protein is a component of one of the largest protein complexes forming MAM. Both PINK1 and Parkin, the products of genes involved in autosomal recessive PD, were found to localize in the MAM and recruit the autophagy machinery to perform mitophagy [165]. Mitophagy is a crucial process of autophagic degradation of damaged mitochondria in lysosomes. A53T αSyn increases mitophagy, presumably as a compensatory mechanism to remove defective mitochondria [166]. Conversely, αSyn overexpression represses PINK/Parkin-mediated mitophagy, which exacerbates mitochondrial damage by the accumulation of defective mitochondria [167,168]. In addition, αSyn impairs autophagy by significantly downregulating light chain 3A (LC3), the loss of which can promote αSyn spreading [169]. Interestingly, cardiolipin under stress conditions can migrate from IMM to OMM and bind A53T and E46K αSyn, which increases LC3 recruitment and triggers extensive mitophagy [170].

In addition to mitophagy, αSyn also influences other key mitochondrial life cycle processes, including mitochondrial fusion, fission, and biogenesis. Mitochondrial fusion is a process of amalgamation of two mitochondria into an elongated one, and mitochondrial fission is a process of division of one mitochondrion into two. An excess of αSyn shifts the mitochondrial fission–fusion balance towards mitochondrial fragmentation. αSyn binds to the curved membranes of OMM, thus inhibiting mitochondrial fusion and fostering fission [171]. αSyn can promote mitochondrial fragmentation directly via interaction with membranes, which is subsequently followed by a decline in respiration and neuronal cell death [172]. As mitochondria undergo axonal transport anterogradely toward the synapse and retrogradely toward the perikaryon, αSyn also seems to impinge upon this process [138]. Another important event impeded by αSyn is mitochondrial biogenesis, the key regulator of which is PGC-1α [173]. αSyn impairs mitochondrial biogenesis by reducing PGC-1α protein levels [168]. Also, A30P αSyn downregulates expression of both *PGC-1α* genes (*RG-PGC-1α* and *CNS-PGC-1α*), which in turn exacerbates αSyn oligomerization and toxicity, resulting in a vicious cycle of reciprocal interactions [173].

### 3.2. αSyn-Induced Disturbances in the Ubiquitin-Proteasome System (UPS) and Autophagy-Lysosomal Pathway (ALP)

αSyn is cleared primarily by ALP and UPS, depending on its concentration and aggregation form, and both pathways can complement one another. UPS embraces ubiquitination and subsequent proteolysis of short-lived, misfolded, and damaged proteins. ALP targets long-lived, aggregated proteins and damaged organelles for degradation by two main processes: macroautophagy (MA) and chaperone-mediated autophagy (CMA). In MA, substrates are sequestered within double-membrane vesicles (autophagosomes) that subsequently fuse with lysosomes into autophagolysosomes, in which the contents are degraded. This process is orchestrated by autophagy-related proteins (ATGs) such as Beclin-1 and the autophagy-related protein microtubule-associated protein 1 light chain 3 (LC3) family. The ubiquitin-binding protein p62, in conjunction with LC3, aids in recognizing and transporting protein aggregates to autophagosomes [174]. In CMA, proteins that contain a KFERQ-like motif (VKKDQ) in their sequence, such as αSyn, are specifically recognized by the chaperone heat-shock cognate 70 kDa (Hsc70). Recognized proteins are in turn targeted and delivered to the lysosome by the lysosomal-associated membrane protein 2A (LAMP2A) [175]. UPS and CMA primarily handle the clearance of normal and soluble αSyn. Meanwhile, MA plays a role in clearing mutated, modified, and aggregated αSyn [176]. In fact, post-mortem studies have confirmed the presence of UPS, CMA, and MA alterations in PD pathology, as exemplified by decreased levels of LAMP2A, Hsp70, catalytic enzymes, proteasome components, and lysosomes, as well as increased levels of LC3-II [177,178,179].

Genetic and mechanistic studies have provided a compelling piece of evidence linking αSyn pathology with dysfunction in UPS and ALP. Point mutations in the *PRKN* gene, which are characteristic of autosomal recessive juvenile PD, impair αSyn labeling with ubiquitin and, consequently, degradation by UPS [180]. Loss-of-function mutations in the *GBA1* gene are a significant risk factor for developing PD. *GBA1* encodes β-glucocerebrosidase (GCase), an enzyme responsible for degrading glucosylceramide into ceramide and glucose. Mutations in the *GBA1* gene reduce the enzymatic activity of GCase, which leads to the accumulation of its substrates—glucocerebroside and αSyn. Their cross-talk leads to the exacerbation of lysosomal dysfunction and the accumulation of αSyn. Impaired αSyn degradation due to defective lysosomal and autophagic processes can result in increased release of αSyn by exosomes, thereby facilitating the spread of αSyn pathology in the brain [181]. Gain-of-function mutations in *LRRK2* are implicated in the autosomal dominant form of PD. Similar to *GBA1*, mutant *LRRK2* contributes to the aggregation of αSyn, the exocytosis of toxic species, and their subsequent spread to other cells [182]. Some other genes involved in PD pathology (*ATP13A2* and *VPS35*) are known to cause ALP dysfunction [183,184]. Mutations in the *VPS35* gene have been detected in patients with autosomal dominant PD. *VPS35* deficiency or mutation (D620N) leads to αSyn accumulation and reduced CMA-mediated αSyn degradation due to a decrease in LAMP2A [184]. *ATP13A2* participates in transmembrane lysosomal transport and seems to have a protective role in αSyn toxicity, whereas its deficiency is characteristic of familial PD. The *ATP13A2* mutation leads to holistic lysosomal alteration, αSyn overexpression, spreading, and enhanced toxicity [185,186].

Both A30P and A53T mutant forms of αSyn impair the function of the 20S/26S proteasome, resulting in further αSyn accumulation and subsequent cell death [187]. αSyn oligomers have been shown to inhibit the catalytic proteasome subunit 20S, partially via interaction with the regulatory 19S subunit, which occurs preferentially in DA neurons [188]. Interestingly, this proteasome inhibition is not exclusive to mutant αSyn, as it can also be induced by WT and modified αSyn [189]. For example, an increased level of pS129-αSyn decreases the abundance of the UPS machinery, which further enhances αSyn toxicity [190]. The proper functioning of CMA is also compromised by mutant or modified αSyn. Mutant and DA-modified forms of αSyn bind to LAMP2A with high affinity, leading to blockage of CMA-mediated degradation of αSyn and other substrates of this pathway [175,191]. Consistently, larger aggregates require ALP involvement due to UPS and CMA inhibition. However, αSyn may compromise its own degradation and autophagy function; as it cannot be efficiently degraded, the loading and clearance of cargo is significantly impaired, and this promotes the accumulation of proteins (including αSyn). This creates a vicious cycle in which the accumulation of proteins compromises their removal, which intensifies further accumulation and promotes neurodegeneration. Overexpression of WT αSyn was shown to inhibit both ALP pathways, mainly via mislocalization of autophagy-related protein 9 (ATG9) and a decrease in GCase activity [192,193]. αSyn was also shown to alter MA by inhibiting the Rab1 pathway and high mobility group box 1 (HMGB1)-Beclin1 complex formation, which are important mediators of autophagosome maturation and recruitment [192,194]. Dysregulation of other key autophagic molecules like LC3, autophagy-related protein 7 (ATG7), or mammalian target of rapamycin (mTOR) is also a characteristic feature of α-synucleinopathies [195].

DA cell death follows disrupted lysosomal integrity and clearance due to mitochondrial-induced oxidative stress, which causes abnormal lysosomal membrane permeabilization [179]. Moreover, αSyn overexpression disrupts vesicular trafficking between the ER and Golgi apparatus (GA), which in turn impairs hydrolase trafficking and decreases lysosomal function [196]. A recent study has also shown that αSyn overexpression reduces autophagy by compromising the fusion of autophagosomes with lysosomes due to a decrease in the level of SNAP29, a SNARE-family protein that navigates autophagolysosome formation [197]. Research advances continue to unveil a growing number of proteins and miRNA molecules as significant markers of autophagy dysfunction. It is crucial to increase our understanding of molecular factors implicated in PD pathogenesis, as it introduces a broader spectrum of potential targets for disease-modifying therapies [198].

### 3.3. Endoplasmic Reticulum (ER)/Golgi Damage and Unfolded Protein Response (UPR) Signaling Pathway

Under physiological conditions, cellular chaperones in the cytosol and endoplasmic reticulum (ER) oversee the correct folding of newly synthesized proteins. Quality control mechanisms identify misfolded proteins for degradation through the UPS and ALP. This process, known as proteostasis, is crucial for maintaining the proper function of highly sensitive DA neurons and preventing abnormal protein aggregation. However, in the course of PD, proteostasis is significantly disrupted. Accumulation of cargo in the ER machinery leads to protein misfolding, disruption in calcium balance, and axonal transport, which ultimately triggers ER stress. This initiates the unfolded protein response (UPR), a signaling cascade that enhances protein folding, quality control mechanisms, and degradation pathways, but, in cases of irreversible damage, it can also trigger apoptosis [199]. UPR aims to restore proteostasis by blocking global ER protein synthesis, upregulating chaperones, inducing ER-associated degradation (ERAD), and ultimately activating autophagy or apoptosis. The aforementioned events are induced by the three UPR sensors: IRE1 (inositol-requiring protein 1), PERK (protein kinase RNA-like ER kinase), and ATF6 (activating transcription factor 6). Under physiological conditions, these sensors form a complex with BiP (binding immunoglobin protein; also known as GRP78—glucose-regulated protein 78) and remain inactive. However, binding of BiP to aberrant proteins results in dissociation of the complex and subsequent activation of the UPR pathways [200]. Evidence for ER stress and UPR activation in WT, mutant, and overexpressed aSyn-induced damage has been confirmed in multiple experimental models [201,202,203,204]. Furthermore, mutations in various PD-related genes (e.g., *PRKN*, *ATP13A2*, and *GBA1*) have also been associated with the presence of ER stress [183,205,206]. Activated BiP, IRE1/XBP1, and PERK have been demonstrated in the SN of PD patients, and their levels significantly and positively correlated with the increased levels of αSyn [207]. Similarly, increased levels of UPR markers have been observed in MSA [208]. In both α-synucleinopathies, activation of the UPR occurred in the early stage of the disease, which may suggest its important role in the pathology progression [209]. Mechanistically, accumulated αSyn has been found to directly bind to BiP, resulting in activation of the PERK-dependent pathway, proapoptotic changes, and cell death [201]. Upon activation, PERK phosphorylates the α subunit of the eukaryotic translation initiation factor 2α (eIF2α), leading to a temporary inhibition of protein synthesis and upregulation of target genes that encode for ER chaperones and ERAD components. However, if this adaptive response is insufficient to resolve ER stress, cells undergo programmed cell death through activation of the PERK-dependent, pro-apoptotic C/EBP-homologous protein (CHOP) [200]. αSyn not only induces ER stress but also hinders the positive effects of the UPR by interfering with the ERAD pathway. This results in the buildup of substrates and, eventually, CHOP-mediated apoptosis [210,211]. αSyn accumulation due to *SNCA* triplication induces IRE1 activation, downregulation of antiapoptotic Bcl2 protein, and upregulation of proapoptotic BIM and CHOP proteins; this results in extensive autophagic and apoptotic cell death [204]. Overexpression of both WT and mutant forms of αSyn can induce a late UPR event, *XBP1* mRNA splicing (related to the IRE1 pathway), without the involvement of the three stress sensors [212]. Interestingly, soluble oligomers are potent inducers of *XBP1* mRNA splicing in contrast to monomers or fibrils, which points out the toxic nature of these species [213]. αSyn also inhibits ATF6, a protective branch of UPR, both directly via physical interaction and indirectly via restriction of COPII (coat protein complex II) vesicles [214]. COPII is a protein complex that facilitates the formation of vesicles involved in ER-GA protein transport, and it is necessary for ATF6 activation [215]. Furthermore, αSyn itself suppresses the trafficking of proteins from the ER to the GA, which affects protein maturation [216]. Collectively, aberrant αSyn contributes to the holistic impairment of cellular proteostasis, from mRNA translation to the degradation of mature proteins. Interestingly, new research has found that the influence of αSyn extends beyond this cycle. It has been reported that αSyn interacts with a set of RNA-binding proteins involved in mRNA-decapping and degradation, thereby affecting mRNA stability in the cytosol. αSyn accumulation in PD patients’ brains perturbs mRNA metabolism, which gives a broader perspective on protein expression abnormalities in PD [217].

### 3.4. Nuclear Dysfunction

Although αSyn was given its name due to its synaptic (syn-) and nuclear (-nuclein) localization [218], its function in the nucleus remains poorly understood. However, various studies have proposed that αSyn could interact directly with the DNA double helix, histones, and other nuclear proteins, participating in both genetic and epigenetic regulation of gene expression [219,220,221,222,223,224]. Initial evidence of αSyn potential pathogenic impact on the cell nucleus arose from postmortem observations of αSyn nuclear inclusions in neurons and glial cells in patients with MSA [225]. This phenomenon has been further confirmed in patients with PD and LBD, implying a potentially broader involvement of nuclear aggregates across various α-synucleinopathies [226,227]. Notably, αSyn nuclear localization is highly dependent on S129 phosphorylation, as pS129-αSyn is abundantly and predominantly localized within the nucleus and forms inclusions. pS129 facilitates αSyn nuclear import and retention, which disrupts DNA stabilization and repair [226,227,228]. Interestingly, inoculation of αSyn pre-formed fibrils (PFF) into the mouse brain leads to the formation of the pS129-αSyn inclusions, subsequent spreading across neurons, and neuronal death [229]. The above results indicate the potential role of pS129-αSyn and nuclear dysregulation in driving α-synucleinopathies. However, the question of αSyn transport into the cell nucleus remains enigmatic. It seems that monomeric species, due to their size, can freely pass through nuclear pore complexes by passive diffusion [230]. However, larger species may require an active translocation mechanism, potentially through interaction with tripartite motif-containing 28 (TRIM28) or ras-related nuclear protein (RAN) [231,232]. Once inside the nucleus, αSyn can modulate DNA physical properties: monomeric αSyn increases DNA persistence length (stiffening) via electrostatic interactions, while C-truncated αSyn induces distinct DNA changes (e.g., extension and compaction), depending on truncation residue [219,233]. Endogenous WT αSyn colocalizes with DNA damage response components and facilitates a non-homologous end-joining reaction, thus protecting DNA from double-strand breaks (DSBs). However, the protection is abolished under cytosolic αSyn aggregation [220].

WT and pS129-αSyn bind to the tails of both core and linker histones, with the phosphorylated species exhibiting a higher affinity [224]. The αSyn mutant species (A30P, A53T, and G51D) also displayed greater nuclear localization than WT and increased toxicity [223,234]. One proposed mechanism for this toxicity is based upon the direct association of mutant αSyn with histone 3 (H3), which inhibits its acetylation, reduces protective gene expression, and promotes cell death [223]. Indeed, it has been found that the levels of acetylated histone H3 are markedly reduced in the SN of PD patients [235]. In addition to histones, αSyn is also thought to interact with histone-modifying proteins such as histone acetyltransferases (HATs), histone deacetylases (HDAC), and histone methyltransferases (HMTs). HATs are enzymes that add acetyl groups to the lysine residues of histone proteins. Acetylated histones electrostatically repel DNA, leading to chromatin relaxation, which enables attachment of the transcription apparatus and activation of the transcription. Conversely, HDACs remove acetyl groups, diminishing electrostatic repulsion and allowing histones to wrap the DNA more tightly, which suppresses transcription [236]. Overexpressed cytosolic αSyn decreases the HAT activity of the p300 protein, which reduces histone acetylation in nigral DA neurons [120]. Moreover, the A53T mutant αSyn significantly decreases the level of histone H3 acetylation by interacting with transcriptional adapter 2-alpha (TADA2a), a component of the HAT complex p300/CBP [235]. The A53T mutant αSyn interacts with HDAC4, causing its nuclear retention, which in turn leads to altered neuronal gene expression and promotes apoptosis in DA neurons [237]. Besides histones, HDACs can also perform deacetylation on non-histone proteins, such as αSyn. As mentioned before, acetylation of αSyn stabilizes its structure and prevents spontaneous aggregation. Sirtuin 2 (SIRT2), a member of the HDAC family, can directly interact with αSyn and perform its deacetylation, which exacerbates αSyn aggregation and toxicity [238]. Inhibition of HDACs has been proposed as a potential therapeutic strategy against α-synucleinopathy. For instance, sodium butyrate, the HDAC inhibitor, has been shown to protect DA neurons by upregulating genes responsible for DNA repair, thereby reversing damage caused by αSyn [239]. In the context of histone methyltransferases, αSyn overexpression influences histone 3 methylation patterns by recruiting euchromatic histone-lysine N-methyltransferase 2 (EHMT2). This results in elevated levels of histone-H3 lysine-9 (H3K9) dimethylation within the SNAP25 promoter, which affects SNARE complex assembly and contributes to synaptic dysfunction [240].

A genome-wide study conducted on brain and blood samples from PD patients revealed distinct DNA methylation patterns, which might facilitate the development of new tools for molecular diagnosis of the disease [241]. Further research has found that αSyn interacts with DNA-modifying proteins such as DNMTs (DNA methyltransferases). DNMTs represent a class of enzymes that are involved in the transfer of methyl groups to CpG sites of DNA. It has been reported that αSyn affects the DNA methylation process by sequestering the DNMT1 and RAN/DNMT3A axes [232,242]. αSyn-induced demethylation of *SNCA* intron 1 increases the expression of αSyn, as evidenced by brain specimens from patients with α-synucleinopathy [243] (Figure 2).

## 4. αSyn Impact on the Extraneuronal Space

### 4.1. Seeding and Propagation of αSynuclein

In addition to its intracellular functions, αSyn can be secreted by neurons and spread to surrounding cells through various mechanisms. αSyn is believed to spread in a prion-like manner (seeding), as evidenced by fetal mesencephalic DA neurons grafted into the SN of PD patients, which subsequently developed αSyn-positive inclusions. This phenomenon suggests that αSyn has the ability to transfer from diseased neurons to healthy neurons, which contributes to the progressive spread of pathology in PD [244]. It has been proven by multiple studies that exposure of neurons to extracellular preformed fibrils (PPFs) results in their endocytosis and recruitment of soluble endogenous αSyn to form intracellular insoluble inclusions, resembling LBs and LNs (secondary nucleation). Pathology is initially detected in the axons and subsequently propagates into perikaryons, where fibrils undergo gradual maturation. The pathology then propagates to neighboring cells and further spreads to distant anatomically and functionally connected regions. This results in the degeneration of inclusion-bearing neurons and progressive impairment in neuronal network function, ultimately leading to ubiquitous neuronal cell loss and brain atrophy—a macroscopic hallmark of PD [245,246,247,248,249].

Substantial levels of soluble αSyn oligomers have been detected in the plasma and CSF of patients with α-synucleinopathies, which further confirms the spreading of oligomeric species [250,251]. In the CNS, αSyn level depends on the balance between αSyn synthesis and clearance, which can be disturbed by abnormal aggregation processes. Clearance of various forms of αSyn in the extracellular space (ECS) was shown to be mediated by chaperones (mainly heat shock proteins; Hsps) and direct proteolysis by extracellular proteases (plasmin, neurosin, or MMPs) [252]. Pathology is induced when the level of chaperones or proteolytic enzymes is significantly lower than the level of pathological aggregates or when cleavage occurs at specific sites that favor aggregation. Propagation of αSyn from cell to cell happens when the protein is released to the ECS due to leakage from a dead cell or release from a living cell overloaded with αSyn. The latter includes several mechanisms: direct membrane penetration, endocytosis, exocytosis (of synaptic vesicles or exosomes), axonal transport, trans-synaptic spread, and receptor-dependent uptake [253]. Uptake of extracellular αSyn oligomers by donor cells is mediated by several cell surface receptors, including the transmembrane protein lymphocyte-activation gene 3 (LAG3), Aβ precursor-like protein 1 (APLP1), toll-like receptor 2 (TLR-2), heparan sulfate proteoglycans (HSPGs), neurexin 1, and the gap junction protein connexin-32 (Cx32) [254,255]. Exocytosis takes place upon cellular dysfunction, and αSyn may be secreted either by a non-classical exocytic pathway or in a calcium-dependent manner by exosomes [256]. These mechanisms constitute part of the cellular quality control system to dispose of damaged and harmful proteins. Importantly, the spreading involves not only neural but also glial cells, as microglia may also secrete oligomers by exosomes and induce αSyn accumulation in neurons [257]. Once inside the neuron, αSyn aggregates are targeted to lysosomes and transported along the axons to adjacent cells or, further, to the interconnected brain regions. Interestingly, αSyn fibrils, when packed in lysosomes, can be transferred from one cell to another without coming into contact with the ECS. This transfer occurs through tunneling nanotubes, which are intercellular channels that connect two adjacent cells [258]. Extracellular αSyn triggers neurotoxicity, inflammation, and the seed formation of new aggregates in neighboring cells [259]. It has recently been proposed that high-molecular-weight αSyn oligomers exhibit high neurotoxicity by directly damaging plasma membrane integrity and inducing the extrinsic apoptotic pathway, which highlights their potential as a plausible target for the development of PD-modifying therapies [259].

Once αSyn seeds reach the target cells, they act as nuclei and promote further elongation, exerting deleterious effects on the recipient cells [260]. Nucleation and/or seeding of new aggregates can be accelerated by αSyn overexpression (*SNCA* multiplications), the addition of preformed nuclei (seeds and prions), or exogenous surfaces (microbes and nanoparticles) [253,261]. As exogenous fibers bind to membranes, they drive seeding pathology by providing nucleation sites for endogenous αSyn. This leads to the depletion and loss of function of monomers that are sequestered into amyloid aggregates [252]. On the other hand, toxic oligomers may be released from the assemblies and thereby induce pathology [262]. As not all types of oligomer strains were found to induce pathology, it has been proposed that the seeding effect and selectivity towards a specific cell type may rely on certain conformations [263]. Off-target αSyn oligomers and protofibrils are generally thought to be responsible for the seeding of the disease, whereas on-target oligomers are responsible for fibril formation. Also, the diversity of oligomers is potentiated by their ability to convert their structure from one form into another based on environmental conditions [264]. Furthermore, self-templating or cross-seeding of WT or mutant αSyn modulates not only the elongation rate but also the structure of the growing fibrils, which gives rise to conformationally different strains [265]. The cross-seeding effect between αSyn and other amyloidogenic proteins (Aβ, tau) has also been observed. Aβ plaques trigger the accumulation of αSyn in synaptic endings and cause microtubule disassembly. This leads to interactions between mislocalized αSyn and tau proteins. Aβ plaques amplify αSyn seeding, thereby fostering cross-seeding effects on tau [266]. The intrinsic interplay between those pathogenic aggregates is still poorly understood. Nevertheless, considering the heterogeneous characteristics of proteinopathies, cross-seeding may hold relevance in neurodegenerative disease research [267].

According to the Braak staging theory, αSyn propagates to the subsequent brain regions in a specific order. αSyn pathology may have a beginning in the synapses of the PNS of the gut and then, over time, be transmitted to the CNS by the vagus nerve via retrograde axonal transport. The dorsal motor nucleus of the vagus nerve as well as the olfactory bulb and related pathways are the first two areas to exhibit Lewy pathology (stages 1 and 2) [268]. This is in accordance with the fact that many patients experience peripheral, non-motor symptoms years before the disease’s onset. Moreover, patients who underwent appendectomy or vagotomy have an apparently lower risk of developing PD [269,270]. The vagus-dependent spread of pathology has also been confirmed in a mouse model. αSyn PFFs inoculation into the stomach and duodenum of the WT mouse resulted in the development of pS129-αSyn inclusions, first in the dorsal motor nucleus of the vagus nerve, then in the locus coeruleus, amygdala, SN, and finally in the prefrontal cortex, reflecting the Braak staging of PD. This was associated with DA neuron loss, motor dysfunction, and cognitive impairment. Truncal vagotomy abolished the gut-to-brain spread of αSyn and protected against neurodegeneration, which further supports the theory that αSyn pathology may originate from the peripheral nervous system [271]. On the other hand, a recent study in non-human primates has cast doubt upon the involvement of the vagus nerve, pointing to blood circulation as a possible pathway for the bidirectional transport of αSyn between the ENS and CNS [272]. Yet there is a great deal of uncertainty about the connectivity between the ENS and CNS. According to recent reports, unbalanced gut microflora can modulate the host immune response and potentially contribute to the systemic inflammation seen in PD patients [273,274]. The microbiome–gut–brain axis adds another layer of complexity to PD research, and the exact link between the microbiome and the heterogeneity of PD clinical phenotypes has yet to be determined.

### 4.2. Glial Interplay Contributes to Neuroinflammatory Response and Aggravation of αSyn-Induced Damage

#### 4.2.1. Reactive Microglia and Adaptive Immune Response

Neuroinflammation is inherently associated with neurodegenerative diseases. Both innate and adaptive immune systems are implicated in the neurodegenerative process in PD. The key players of the innate immune response in the CNS constitute microglia, resident macrophages of the brain, that are densely distributed in the SNpc and striatum [275]. The first reports on the involvement of microglia in PD came from observations of a reactive microglia phenotype in the postmortem tissue of PD patients, the amount of which appeared to increase with neurodegeneration throughout the nigrostriatal pathway [276,277]. During activation and accumulation, microglia release high levels of pro-inflammatory cytokines, facilitating the recruitment of lymphocytes—components of the adaptive immune response. Indeed, elevated levels of cytokines, including interleukins (ILs: IL-1β, IL-2, and IL6), interferon γ (IFNγ), and tumor necrosis factor α (TNFα), have been detected in the CSF and serum of PD patients [278]. In the SNpc of PD patients, increased infiltration of both helper (CD4+) and cytotoxic (CD8+) T lymphocytes has been observed [279]. Moreover, neuroinflammation weakens the blood–brain barrier and makes it more permeable to peripheral immune cells, which further amplifies the inflammatory response [280]. Interestingly, T-cell-induced responses may significantly differ between individuals, as various strains of αSyn present distinct polypeptide chains on their surface [281]. Induction of ROS and iNOS by an oxidative burst of microglia may also facilitate PTMs of αSyn (oxidation and nitration) and thus promote amyloid formation [282]. αSyn nitrated at all four tyrosine residues is able to escape immune tolerance and generate a deleterious T-cell response towards DA neurons [283]. In addition to PD, the presence of activated microglia and T-cells, as well as increased levels of pro-inflammatory cytokines, are well documented in other α-synucleinopathy cases [284,285].

Microglia express pattern recognition receptors (PRRs) that detect pathogen-associated molecular patterns (PAMPs) or tissue damage-associated molecular patterns (DAMPs). Aggregated extracellular αSyn can act as a DAMP, enabling microglia to recognize it through activation of specific PRRs such as TLRs. Oligomeric αSyn induces activation of TLR-1/TLR-2 signaling, which leads to increased production of ROS and proinflammatory cytokines such as TNF-α, IL-6, and IL-1β [286,287]. TLR-4 activation, on the other hand, can be induced by various αSyn species and contributes to increased microglial phagocytosis, ROS production, and secretion of IL-6, TNF-α, and CXCL1 [288]. Moreover, αSyn fibrils, when opsonized by IgG, can interact with FcγR, which subsequently mediates phagocytosis and proinflammatory NF-κB signaling [289]. It is also worth mentioning that αSyn fibrils, after phagocytosis, activate the NLRP3 (nod-like receptor protein 3) inflammasome and caspase-1, which leads to massive IL-1β secretion [290]. Caspase-1, an enzymatic component of the inflammasome complex, truncates αSyn into pro-aggregatory nuclei, which exacerbates αSyn accumulation and toxicity [291]. In contrast to initiating the deleterious cascade, αSyn also activates the antioxidant transcription factor Nrf2, which reduces the pro-inflammatory response and improves αSyn clearance and neuronal survival [292]. Microglia may also play a neuroprotective role by clearing and degrading αSyn via selective autophagy [18]. Pro-inflammatory microglia are regarded as facilitating agents for αSyn propagation via exosome release [257]. Conversely, under an anti-inflammatory profile induced by IL-4 administration, spreading is significantly reduced [293].

Microglia possess the ability to express MHCII, so they can act as an antigen-presenting cell (APC) for CD4+ T cells. This may lead to an adaptive immune response targeting αSyn. αSyn overexpression increases MHCII production, proliferation, and activation of CD4+ Th cells within SN [294]. Specific peptides derived from αSyn may be displayed by MHCII on the microglia surface and act as epitopes for CD4+ T cell activation. Furthermore, other αSyn-derived peptides can be displayed by MHCI on the surface of DA neurons [295]. It has been previously shown that, upon microglial activation, DA neurons start expressing MHCI, which renders them susceptible to cytotoxic CD8+ T cell-induced damage [296]. A recent study examined T cell infiltration in the SNpc at various PD stages, revealing robust CD8+ T cell infiltration at the earliest stage, before αSyn aggregation, suggesting a potential autoimmune origin of PD [297]. Similarly, CD4+ T cell influx occurs in the preclinical phase of PD and is most pronounced at the onset of motor symptoms. Initially, αSyn-specific Th cells are characterized by heterogeneity and secrete both pro- and anti-inflammatory cytokines like IFN-γ, IL-4, IL-5, and IL-10. However, their pro-inflammatory status tends to increase as they approach the time of diagnosis [298]. It has been reported that PD is associated with an elevated ratio of pro-inflammatory Th1 and Th17 cells, coupled with a reduced count of anti-inflammatory Th2 and regulatory T cells, indicating a predisposition towards a pro-inflammatory immune response [299]. This bias towards Th1/Th17 has been further validated, demonstrating elevated frequencies of Th1 cells and increased inflammation markers in the serum of PD patients. Notably, these peripheral markers did not exhibit a significant correlation with microglial activation in the brains of PD patients. This implies that peripheral adaptive immunity may have an indirect influence on microglial activation during the course of neurodegeneration in PD [300].

It is believed that under stress conditions, both innate and adaptive immune systems may act as a double-edged sword by exerting either protective or deleterious effects on the neural environment. In the context of microglial activation, the prevailing theory suggests that, upon differential activation, microglia can adopt either a pro-inflammatory, neurotoxic M1 phenotype or an anti-inflammatory, neuroprotective M2 phenotype. Although this terminology is commonly used in microglia research, recent advances in single-cell or single-nucleus RNA sequencing have enabled the identification of incredible heterogeneity and unique disease-related signatures in these reactive microglia. Collectively, these studies revealed a distinct transcriptional signature of disease-specific microglia activation in the midbrain and striatum of PD patients, characterized by enriched expression of genes responsible for neuroinflammation, phagocytosis, and cell proliferation [301,302]. While the canonical M1/M2 classification may not accurately describe microglial states, it aids in understanding the therapeutic potential of microglial modulation. Multiple studies suggest that microglia, under certain cytokine profiles, play a key role in the modulation of αSyn-induced neurodegeneration and spreading, which can be aggravated by the pro-inflammatory M1 phenotype and alleviated by the anti-inflammatory M2 phenotype. Therefore, the promotion of M2 microglia may provide a novel immunomodulatory therapeutic strategy against PD [303].

#### 4.2.2. Astrocytes

The crosstalk between microglia and astrocytes has been linked to dual modulation of PD progression, as they can act both protectively and harmfully, depending on the type of stimulation. Similar to microglia, astrocytes display a great level of heterogeneity [304]. Nevertheless, the prevailing theory holds that astrocytes may differentiate from inactive A0 into two reactive profiles that mirror microglial M1/M2 activation—neurotoxic A1 and neuroprotective A2 [305]. Microglia play a crucial role in triggering neurotoxic A1 astrocyte reactivity through the release of IL-1α, C1q, and TNF-α, creating a feedback loop of dysregulated inflammation that accelerates the progression of brain injury. Indeed, reactive astrocytes are present in post-mortem tissue in most neurodegenerative diseases, suggesting that they may contribute to neuronal death [19]. Although astrocytes have very little ability to express αSyn, LB-like inclusions have been found in astrocytes from PD patients, which suggests their ability to engulf the extracellular αSyn released by neurons [306]. Astrocytes exposed to extracellular αSyn may undergo both receptor-mediated endocytosis and membrane receptor interactions. While both TLR-2 and TLR-4 are associated with proinflammatory effects, TLR-2 is implicated in αSyn uptake. TLR-2 enhances the cellular uptake of αSyn fibrils while reducing its degradation, and it also triggers neurotoxic proinflammatory responses in astrocytes via the NF-κB pathway. TLR-2 inactivation results in activation of autophagy and increased clearance of αSyn while inhibiting the inflammatory response; this improved neuropathology and mitigated behavioral defects in α-synucleinopathy animal models [307,308,309]. Activation of astrocytes by αSyn oligomers may also occur via the TLR-4-dependent pathway. TLR-4 is the most highly expressed in SNpc and is extremely sensitive to αSyn oligomers, as it is able to recognize them at picomolar concentrations, which reinforces its role in PD initiation. TLR-4 activation induces NFκB nuclear translocation, with subsequent extensive cytokine release (e.g., TNF-α and IL-6) and ROS production, which culminates in neural cell death [288,310]. All forms of αSyn activate astrocytes and upregulate oxidants and cytokines, but, in particular, the long-term storage of oligomers induces mitochondrial dysfunction and oxidative stress by increasing extracellular hydrogen peroxide production [311]. On the other hand, αSyn PFFs can activate astrocytes through necroptotic kinases-dependent activation of NF-κB signaling, shifting the phenotype towards neurotoxic A1 [312]. Interestingly, blockage of PFFs-induced M1 and A1 conversion has been shown to prolong survival and reduce neuronal impairment in a PD mouse model, underscoring the pivotal role of these phenotypes in PD progression [313].

In addition to neuroinflammation, astrocytes contribute to the uptake and clearance of extracellular αSyn. After internalization, astrocytes begin to degrade both αSyn oligomers and fibrils via the lysosomal pathway, acting as neuroprotective scavengers that limit the formation of the inclusions. However, under an excessive burden, lysosomal degradation machinery becomes overloaded, resulting in incomplete digestion of the inclusions. Subsequently, autophagy is suppressed, which causes severe damage to mitochondria and sensitizes astrocytes to apoptosis; this results in the exacerbation of neurodegeneration [311,314,315]. Since astrocytes transfer αSyn among neurons and other glia, it has been concluded that astrocytes can act as spreading agents. However, the extent of deposit clearance in astrocytes is greater than spreading, especially when assisted by microglia; this emphasizes the pivotal role of glial interplay in neuronal protection [316,317].

Overall, the role of the astrocyte and microglia response to αSyn uptake is complex, and further studies are needed to clarify its exact role in PD. Nonetheless, it is believed that the initial activation of glial cells in response to toxic species promotes phagocytosis and clearance, thereby preventing αSyn transmission and protecting against the development of pathology. However, as glial cells accumulate damage associated with αSyn overload, they may become predisposed to adopt a harmful, disease-accelerating phenotype at later stages [318].

#### 4.2.3. Oligodendrocytes

While the involvement of microglia and astrocytes in PD is well documented, there is limited evidence regarding the impact of αSyn on oligodendrocytes in the pathogenesis of PD. The presence of αSyn aggregates in oligodendrocytes in the form of glial cytoplasmic inclusions (GCIs) constitutes a pathological hallmark of MSA. The origin of αSyn in GCI remains enigmatic, with competing studies proposing either internalization of neuronally secreted αSyn by oligodendrocytes or increased expression and decreased degradation of oligodendroglial αSyn [319,320,321]. In view of recent evidence on differences in the structure, morphology, and toxicity of αSyn aggregates across different α-synucleinopathies [281], a direct translation of αSyn impact on oligodendrocytes from MSA to PD may not be adequate. In contrast to MSA, in PD, αSyn pathology in oligodendrocytes is sparse and appears late in the course of the disease, suggesting that oligodendrocytes do not play a leading role in PD but might be involved in late disease progression [322]. However, recent studies using high-throughput techniques have shed light on the potentially crucial role of oligodendrocytes in PD. It has been demonstrated that PD-related genes are specifically upregulated in the oligodendroglial lineage cells, even in the earliest stages of PD, suggesting that changes in oligodendrocytes may precede the onset of pathology [323,324]. Recent studies have shown that oligodendrocytes in PD have disease-specific molecular signatures characterized by impaired maturation, protein folding stress, and inflammatory reprogramming [325,326]. In fact, the loss of mature myelinating oligodendrocytes and a significant reduction in myelin content, particularly in connections originating from the SN, have been observed in postmortem studies of PD patients. [301,327]. The extent of myelination has been previously posited as a critical determinant of neuronal vulnerability to αSyn-induced damage in PD [328]. Although overt impairment of oligodendroglial and myelin homeostasis and associated white matter lesions are not as pronounced in PD as they are in MSA [329], the seemingly subtle response of oligodendrocytes to toxic αSyn aggregates may in fact tune the neuronal microenvironment, thereby playing a significant role in PD pathogenesis. Nevertheless, further research is needed to elucidate the impact of oligodendroglial lineage cell disruption on PD progression and identify potential therapeutic targets within oligodendroglial-related molecules or functional pathways.

#### 4.2.4. Excitotoxicity

Excitotoxicity is a phenomenon characterized by glutamate accumulation at the synapse and thus overload in N-methyl-D-aspartate (NMDA) and α-amino-3-hydroxy-5-methylisoxazole-4-propionic acid (AMPA) receptor capacity. This in turn leads to an intracellular influx of Ca^2+^ and Na^+^, oxidative stress, apoptotic cell death, and escalation of the neuroinflammatory response [330]. Astrocytes are responsible for maintaining glutamate homeostasis by regulating the balance between glutamine uptake and release [331]. A recent study has identified a distinct subpopulation of specialized astrocytes, characterized by a unique molecular signature reminiscent of glutamatergic synapses. These astrocytes communicate with the environment through glutamatergic gliotransmission. Importantly, they play a crucial role in modulating glutamatergic synaptic transmission onto nigral DA neurons, contributing to the regulation of nigrostriatal circuitry function [332]. This may have significant implications for PD and warrant investigation in forthcoming studies.

Studies have shown that astrocytic overexpression of A53T αSyn significantly reduces the levels of the crucial glutamine transporters—excitatory amino acid transporters 1 and 2 (EAAT1 and EAAT2). This results in impaired glutamine uptake, excitotoxicity, severe neuroinflammation, and microgliosis in the brainstem [333]. Furthermore, αSyn monomers and fibrils have been shown to directly stimulate glutamate release from presynaptic terminals, potentially contributing to excitotoxicity [334]. There is a growing recognition of the key role of NMDA receptor dysfunction and impaired long-term potentiation (LTP) in a variety of CNS conditions, from childhood neurodevelopmental disorders to late-onset neurodegenerative diseases [335,336]. Interestingly, αSyn oligomers may activate NMDA receptors, which increases basal synaptic transmission and impairs LTP. The synaptic damage is subsequently triggered by disruptions in calcium homeostasis and membrane integrity. This leads to deterioration in essential neurophysiological functions, like learning and memory [337]. Furthermore, the enhancement of AMPA-mediated transmission by αSyn oligomers disturbs intracellular Ca^2+^ homeostasis, contributing to excitotoxic neuronal death [338]. αSyn may also sensitize NMDA receptors to glutamate excitotoxicity via increased phosphorylation of its subunits (NR1 and NR2B) in the SN and striatum [339]. Altogether, these findings suggest that αSyn can exert an impact on glutamate clearance, release, and transmission, both in astrocytes and neurons. This indicates the essential role of excitotoxicity in PD-related neurodegeneration.

## 5. Summary and Perspective

As life expectancy continues to rise, neurodegenerative diseases present a growing challenge to modern medicine. From 1990 to 2016, the global prevalence of PD increased by 74.3%, and the upward trend continues all the time, intensifying the socioeconomic burden of the disease [340]. The heterogenous clinical manifestations as well as the multifactorial background of PD suggest that the pathogenesis of PD can vary from person to person. This not only points to the importance of personalizing therapy to meet a patient’s unique needs but also underscores the necessity of developing specific diagnostic tests and biomarkers for PD; such tools could allow for the identification of the patient’s molecular signature and monitoring of the effects of the targeted therapy. Despite extensive research, a definitive cure for PD is difficult to achieve. While current therapies offer relief from early motor symptoms, their effectiveness declines as the disease progresses. Currently, most drugs target the pre- or postsynaptic regulation of DA nigrostriatal transmission, aiming to reverse the effects of DA insufficiency. These drugs include not only the gold standard L-DOPA but also catechol-O-methyltransferase inhibitors, monoamine oxidase type B inhibitors, and DA agonists. Other drugs used in PD largely aim to abate the adverse effects of the former. Understanding the electrophysiology of the basal nuclei circuitry allowed the discovery of deep-brain stimulation (DBS) [341]. Although it has significantly improved the quality of life of PD patients, especially drug-resistant ones, it still does not constitute a disease-modifying therapy leading to a cure. In part, the development of neuroprotective or neurorestorative therapies has failed due to an incomplete understanding of the pathophysiological mechanisms involved in PD progression and the limited identification of drugs that can treat the known pathways. Discovering new therapeutic agents that target abnormal αSyn species and prevent their aggregation requires a comprehensive understanding of the initial triggers of αSyn aggregation, mechanisms underlying cellular damage, and how aggregation spreads in neurons and throughout the brain.

In this review, we have presented the current state-of-the-art regarding the αSyn structure; its putative role in synaptic regulation; factors that contribute to the aggregation, formation, and spread of toxic species; as well as the deleterious effects of αSyn on neuronal and glial function. It is widely believed that the smaller oligomers are the main pathogenic factors in PD. Importantly, αSyn oligomers-induced damage potentiates their further production, which results in a vicious cycle. Furthermore, toxic oligomers activate glial cells, triggering neuroinflammation that recruits the peripheral immune system. The neuroinflammatory response in PD involves an interplay between the innate and adaptive immune systems. This places PD at the crossroads of classic neuroinflammatory diseases such as multiple sclerosis, in which the adaptive system is predominantly involved, and other neurodegenerative diseases such as Alzheimer’s disease, in which the innate response prevails [342]. Lastly, the role of astrocytes in controlling glutamatergic transmission and the phenomenon of excitotoxicity that accompanies neurodegeneration may represent further aspects of PD pathology to be unraveled in upcoming studies.

Extensive research has demonstrated the remarkable ubiquity of αSyn and enabled the detection and monitoring of its peripheral deposits. This has translated into the discovery of readily available biomarkers from plasma, blood, CSF, or body tissues such as skin. Undoubtedly, one of the most groundbreaking discoveries regarding PD has been the refinement of SAA, which may become a highly accurate method for molecular PD diagnosis at very early stages of the disease [88]. Indeed, SAA demonstrates high diagnostic performance and remarkable reproducibility in detecting early Parkinson’s disease, with sensitivity ranging from 86 to 96% and specificity from 93 to 100% across different laboratories [343]. In the largest study to date in prodromal and high-risk groups for the development of PD (patients with rapid eye movement sleep behavior disorder or hyposmia), 86% (44 of 51 participants) had positive αSyn SAA [88]. Further progress could pave the way for minimally invasive screening programs that would allow early identification and implementation of therapy for people at risk for PD. Another momentous discovery is the detailed depiction of the structure of αSyn fibrils and the conformational diversity of distinct strains [81,82,83,84,85]. Nevertheless, research on disease-causing oligomers remains challenging due to their instability, transient nature, and high degree of heterogeneity. Accordingly, the high-resolution structure of αSyn oligomers still remains elusive, which warrants further investigation.

Increased knowledge of the molecular mechanisms underlying DA cell death may provide a foundation for the development of new targeted therapies for PD. Targeting the process of αSyn aggregation and the molecular pathways that accompany αSyn-induced damage appears to be a very promising therapeutic strategy against α-synucleinopathies nowadays. Monomer-stabilizing agents and αSyn aggregation inhibitors, such as NPT100-18A, Anle138b, and SynuClean-D, interact with disordered αSyn structure, preventing its β-sheet conversion, self-assembly, and seeding. Targeting proteostasis impairment can be achieved by promoting protein folding, reducing ER stress, and supporting degradation pathways with chaperones, small-molecule inhibitors, or gene therapies. The efficacy of the above-mentioned methods is still being evaluated in preclinical and clinical studies. Finally, modulating mitochondrial pathways, oxidative stress, and nuclear homeostasis adds further evidence of how understanding the pathogenesis of PD can be directly translated into precision medicine approaches. When targeting αSyn, it is crucial to maintain a balance between the selective inactivation of toxic species and the preservation of physiological αSyn involved in regulating neurotransmission. Given the indisputable role of αSyn aggregates in the pathogenesis of PD, anti-aggregation therapies appear to be the most promising strategy, as they may offer a causal treatment for idiopathic PD. In contrast, in familial cases with specific mutations underlying the disease, gene therapy seems to be the most reasonable. However, considering the multifaceted molecular changes in PD pathogenesis and uncertainties about the direct role of αSyn, combination therapy addressing various molecular pathways emerges as the most promising approach for the future [344]. Until then, however, the efficacy and safety of such approaches will have to be thoroughly evaluated in large-scale trials to bring the greatest benefit to patients with the fewest side effects.

## Figures and Tables

**Figure 1 ijms-25-00360-f001:**
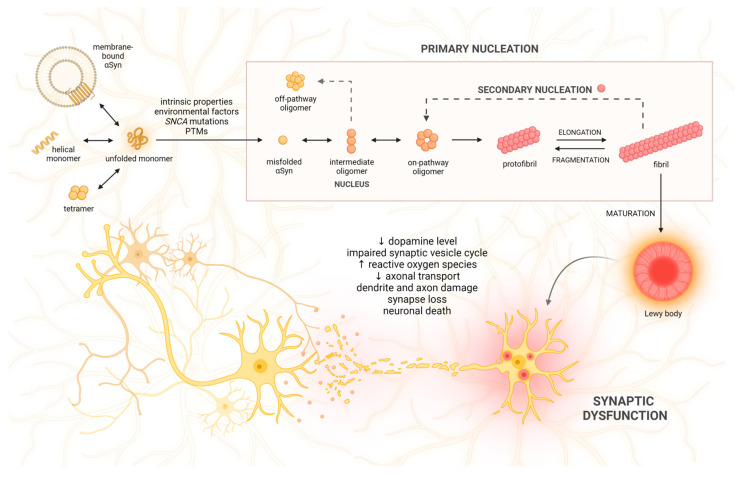
The mechanism of alpha-synuclein (αSyn) aggregation and its effect on synaptic function. Natively unfolded αSyn monomers may take the form of α-helix, tetramer, or membrane-bound multimers. A number of factors, like intrinsic properties of the protein, biological and chemical stimuli, mutations of the αSyn gene (*SNCA*), and post-translational modifications (PTMs), may initiate misfolding of αSyn and aggregation. The primary nucleation process includes the formation of intermediate species (nuclei), which, along with monomer addition, form sequential oligomers, protofibrils, and fibrils. Alternatively, intermediate forms may generate off-pathway oligomers that resist aggregation. Fibrils may undergo fragmentation into smaller aggregates, create secondary seeds, or further assemble into pathological inclusions called Lewy bodies. Aggregated, toxic species of αSyn significantly impair synaptic function, as they interfere with dopamine synthesis, reuptake, and release; affect the synaptic vesicle cycle; generate reactive oxygen species; reduce axonal transport; and eventually contribute to neuronal apoptosis (↑—increased; ↓—decreased).

**Figure 2 ijms-25-00360-f002:**
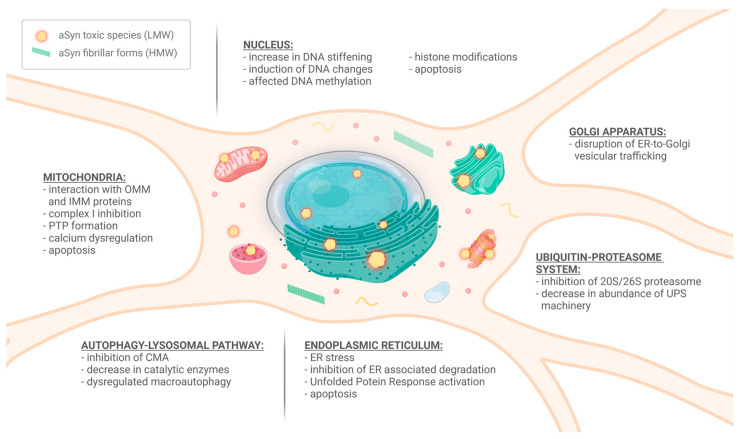
Schematic representation of the effect of alpha-synuclein (αSyn) aggregates on the function of various cellular organelles. While high-molecular-weight (HMW) species of αSyn primarily form inclusions within the cell, the low-molecular-weight (LMW) oligomers are believed to induce damage to different cellular compartments in numerous mechanisms. (OMM—outer mitochondrial membrane; IMM—inner mitochondrial membrane; PTP—permeability transition pore; CMA—chaperone-mediated autophagy; ER—endoplasmic reticulum; UPS—ubiquitin-proteasome system).

## Data Availability

Data sharing is not applicable to this article.

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
