# Peer review of "Alpha-Synuclein Contribution to Neuronal and Glial Damage in Parkinson’s Disease"

_ijms, 2023, doi:10.3390/ijms25010360_

Round 1

Reviewer 1 Report

Comments and Suggestions for Authors

In this paper, Kamil has summarized the Alpha-Synuclein Contribution to Neuronal and Glial Damage in Parkinson’s Disease. It is interesting, but there are some following problems.

1、          When alpha synuclein is engulfed by glial cells, do glial cells play a protective or harmful role, or a combination of both?

2、          How accurate and specific is SAA as an early diagnosis of PD?

3、          From the summary of these treatment approaches, which one do the authors believe is the most promising for success?

Author Response

Response to Reviewer 1 Comments

We are grateful to the Reviewer for the care with which our manuscript was read and for the constructive criticism and valuable suggestion. Taking into consideration the suggestions provided, we present an updated version of the article entitled Alpha-Synuclein Contribution to Neuronal and Glial Damage in Parkinson’s Disease. Essential changes were made to the text of our manuscript according to the comments and suggestions provided. In the revised version of the manuscript all changes made are highlighted in yellow. We confirm that all authors listed on the manuscript concur with the submission in its revised form. We hope that the revisions in the manuscript and our accompanying responses have made the manuscript suitable for publication in International Journal of Molecular Sciences.

Herein, we explain the revisions that were made based on the comments and recommendations.

General remark: In this paper, Kamil has summarized the Alpha-Synuclein Contribution to Neuronal and Glial Damage in Parkinson’s Disease. It is interesting, but there are some following problems.

Response: We would like to thank the Reviewer for the positive feedback.

Point 1: When alpha synuclein is engulfed by glial cells, do glial cells play a protective or harmful role, or a combination of both?

Response 1: We are grateful for the Reviewer’s insightful remark. To address it, we have elaborated on the impact of the engulfed alpha synuclein on glial cells in the according section of the manuscript:

Overall, the role of the astrocyte and microglia response to αSyn uptake is complex, and further studies are needed to clarify its exact role in PD. Nonetheless, it is believed that the initial activation of glial cells in response to toxic species promotes phagocytosis and clearance, thereby preventing αSyn transmission and protecting against the development of pathology. However, as glial cells accumulate damage associated with αSyn overload, they may become predisposed to adopt a harmful, disease-accelerating phenotype at later stages [320].

Point 2: How accurate and specific is SAA as an early diagnosis of PD?

Response 2: We wish to thank the Reviewer for raising this very important question. Accordingly, we have added information on the accuracy and specificity of SAA in terms of early diagnosis of PD, which reads as follows:

(…) Indeed, SAA demonstrates high diagnostic performance and remarkable reproducibility in detecting early Parkinson's disease with sensitivity ranging from 86 to 96% and specificity from 93 to 100% across different laboratories [345]. In the largest study to date in prodromal and high-risk groups for the development of PD (patients with rapid eye movement sleep behavior disorder or hyposmia), 86% (44 of 51 participants) had positive αSyn SAA [89]. (…)

Point 3: From the summary of these treatment approaches, which one do the authors believe is the most promising for success?

Response 3: We thank the Reviewer for this comment and have extended the ‘Summary and Perspective’ section in our concluding remarks on, to the best of our knowledge, the most promising treatment approaches for PD as follows:

(…) When targeting αSyn, it is crucial to maintain a balance between the selective inactivation of toxic species and the preservation of physiological αSyn involved in regulating neurotransmission. Given the indisputable role of αSyn aggregates in the pathogenesis of PD, anti-aggregation therapies appear to be the most promising strategy as they may offer a causal treatment for idiopathic PD. In contrast, in familial cases with specific mutations underlying the disease, gene therapy seems to be the most reasonable. However, considering the multifaceted molecular changes in PD pathogenesis and uncertainties about the direct role of αSyn, combination therapy addressing various molecular pathways emerges as the most promising approach for the future [346]. (…)

Reviewer 2 Report

Comments and Suggestions for Authors

The reviewed work is devoted to the analysis of the role of alpha-synuclein in neuronal and glial pathology in Parkinson's disease (PD). The topic is widely researched and discussed due to the lack of effective therapeutic approaches for the treatment of this disease, and due to the undeniable important role of alpha-synuclein in the pathogenesis PD. In this regard, a sufficient number of review publications on this topic are presented in the scientific literature. Saramowicz and co-authors presented their view on this problem.

Although the title of the manuscript is “Alpha-Synuclein Contribution to Neuronal and Glial Damage in Parkinson’s Disease,” it is not limited to considering only this aspect, but analyzes data on the structure of the protein, its modifications, role in various cellular processes in health and disease, and more. In the abstract, the authors emphasize that “we provide an overview of recent studies emphasizing the multifaceted nature of αSyn and its impact on both neuron and glial cell damage.” Indeed, almost 70% of the works cited in the review were published in the last decade.

Inappropriate and unjustified, in my opinion, is the authors’ brief summary of the material, which has already been presented in detail earlier (Lines 996-1022. “In a nutshell, αSyn is an intrinsically disordered….. neuroinflammation in PD involves an interplay between the innate and 1021 adaptive immune systems”).

The second point concerns the description of the role of alpha-synuclein in glial cells. Microglia and astrocytes are discussed in detail, but there is not even a mention of oligodendrocytes, which makes the discussion of this topic incomplete.

The inscriptions on the pictures should also be made clearer and more readable.

Comments on the Quality of English Language

Minor editing of English language

Author Response

Response to Reviewer 2 Comments

We are grateful to the Reviewer for the care with which our manuscript was read and for the constructive criticism and valuable suggestion. Taking into consideration the suggestions provided, we present an updated version of the article entitled Alpha-Synuclein Contribution to Neuronal and Glial Damage in Parkinson’s Disease. Essential changes were made to the text of our manuscript according to the comments and suggestions provided. In the revised version of the manuscript all changes made are highlighted in yellow. We confirm that all authors listed on the manuscript concur with the submission in its revised form. We hope that the revisions in the manuscript and our accompanying responses have made the manuscript suitable for publication in International Journal of Molecular Sciences.

Herein, we explain the revisions that were made based on the comments and recommendations.

General remarks: Although the title of the manuscript is “Alpha-Synuclein Contribution to Neuronal and Glial Damage in Parkinson’s Disease,” it is not limited to considering only this aspect, but analyzes data on the structure of the protein, its modifications, role in various cellular processes in health and disease, and more. In the abstract, the authors emphasize that “we provide an overview of recent studies emphasizing the multifaceted nature of αSyn and its impact on both neuron and glial cell damage.” Indeed, almost 70% of the works cited in the review were published in the last decade.
Response: We would like to thank the Reviewer for their meticulous analysis and positive feedback.

Point 1: Inappropriate and unjustified, in my opinion, is the authors’ brief summary of the material, which has already been presented in detail earlier (Lines 996-1022. “In a nutshell, αSyn is an intrinsically disordered….. neuroinflammation in PD involves an interplay between the innate and 1021 adaptive immune systems”).

Response 1: We are grateful to the Reviewer for pointing this out. Accordingly, we have shortened the indicated concluding remarks into a more condensed form, so as not to unnecessarily repeat the previously described information:

(…) It is widely believed that the smaller oligomers are the main pathogenic factors in PD. Importantly, αSyn oligomers-induced damage potentiates their further production, which results in a vicious cycle. Furthermore, toxic oligomers activate glial cells, triggering neuroinflammation that recruits the peripheral immune system. (…)

Point 2: The second point concerns the description of the role of alpha-synuclein in glial cells. Microglia and astrocytes are discussed in detail, but there is not even a mention of oligodendrocytes, which makes the discussion of this topic incomplete.

Response 2: We wish to thank the Reviewer for raising this important point. Thanks to this comment, we have now extended the discussion to the role of alpha-synuclein in oligodendrocytes in a new subsection ‘4.2.3 Oligodendrocytes’ as follows:

4.2.3. Oligodendrocytes

While the involvement of microglia and astrocytes in PD is well-documented, there is limited evidence regarding the impact of αSyn on oligodendrocytes in the pathogenesis of PD. The presence of αSyn aggregates in oligodendrocytes in the form of glial cytoplasmic inclusions (GCIs) constitutes a pathological hallmark of MSA. The origin of αSyn in GCI remains enigmatic, with competing studies proposing either internalization of neuronally secreted αSyn by oligodendrocytes or increased expression and decreased degradation of oligodendroglial αSyn [321–323]. In view of recent evidence on differences in the structure, morphology, and toxicity of αSyn aggregates across different α-synucleinopathies [282], a direct translation of αSyn impact on oligodendrocytes from MSA to PD may not be adequate. In contrast to MSA, in PD, αSyn pathology in oligodendrocytes is sparse and appears late in the course of the disease, suggesting that oligodendrocytes do not play a leading role in PD, but might be involved in late disease progression [324]. However, recent studies using high-throughput techniques have shed light on the potentially crucial role of oligodendrocytes in PD. It has been demonstrated that PD-related genes are specifically upregulated in the oligodendroglial lineage cells, even in the earliest stages of PD, suggesting that changes in oligodendrocytes may precede the onset of pathology [325,326]. Recent studies have shown that oligodendrocytes in PD have disease-specific molecular signatures characterized by impaired maturation, protein folding stress and inflammatory reprogramming [327,328]. In fact, the loss of mature myelinating oligodendrocytes and a significant reduction in myelin content, particularly in connections originating from the SN, have been observed in postmortem studies of PD patients. [303,329]. The extent of myelination has been previously posited as a critical determinant of neuronal vulnerability to αSyn-induced damage in PD [330]. Although overt impairment of oligodendroglial and myelin homeostasis and associated white matter lesions are not as pronounced in PD as they are in MSA [331], the seemingly subtle response of oligodendrocytes to toxic αSyn aggregates may in fact tune the neuronal microenvironment, thereby playing a significant role in PD pathogenesis. Nevertheless, further research is needed to elucidate the impact of oligodendroglial lineage cells disruption on PD progression and identify potential therapeutic targets within oligodendroglial-related molecules or functional pathways.

Point 3: The inscriptions on the pictures should also be made clearer and more readable.

Response 3: We thank the Reviewer for their suggestion and we have improved the inscriptions on the pictures. The Reviewer can now appreciate better the clarity of the figures.

Reviewer 3 Report

Comments and Suggestions for Authors

This is an excellent, up-to-date and well written review on PD pathogenic processes.

Manuscript is acceptable in current form, however I have two very minor suggestions.

Line 741: Is it also worth mentioning heparan sulfate proteoglycans (HSPGs) as another cell surface receptor involved in the uptake of αSyn-oligomers .

Line 769: Correct “cross-seeing” to “cross-seeding”.

Author Response

Response to Reviewer 3 Comments

We are grateful to the Reviewer for the care with which our manuscript was read and for the constructive criticism and valuable suggestion. Taking into consideration the suggestions provided, we present an updated version of the article entitled Alpha-Synuclein Contribution to Neuronal and Glial Damage in Parkinson’s Disease. Essential changes were made to the text of our manuscript according to the comments and suggestions provided. In the revised version of the manuscript all changes made are highlighted in yellow. We confirm that all authors listed on the manuscript concur with the submission in its revised form. We hope that the revisions in the manuscript and our accompanying responses have made the manuscript suitable for publication in International Journal of Molecular Sciences.

Herein, we explain the revisions that were made based on the comments and recommendations.

General remark: This is an excellent, up-to-date and well written review on PD pathogenic processes. Manuscript is acceptable in current form, however I have two very minor suggestions.
Response: We wish to thank the Reviewer for the positive feedback and recommendation for publication in IJMS.

Point 1: Line 741: Is it also worth mentioning heparan sulfate proteoglycans (HSPGs) as another cell surface receptor involved in the uptake of αSyn-oligomers .

Response 1: We are grateful for this valuable remark. We added this information in the indicated section of the manuscript, which now reads as follows:

(…) Uptake of extracellular αSyn oligomers by donor cells is mediated by several cell surface receptors, including the transmembrane protein lymphocyte-activation gene 3 (LAG3), Aβ precursor-like protein 1 (APLP1), toll-like receptor 2 (TLR-2), heparan sulfate proteoglycans (HSPGs), neurexin 1, and the gap junction protein connexin-32 (Cx32) [255,256]. (…)

Point 2: Line 769: Correct “cross-seeing” to “cross-seeding”.

Response 2: We thank the Reviewer for pointing this out. Thanks to the Reviewer’s comment, the typo is now corrected in the current version of the manuscript.

(…) Furthermore, self-templating or cross-seeding of WT or mutant αSyn not only modulates the elongation rate, but also the structure of the growing fibrils, which give rise to conformationally different strains [266]. (…)

Reviewer 4 Report

Comments and Suggestions for Authors

This is a well-written review.

However, I think it would be a more excellent paper if you add some references.

Regarding α-synuclein aggregation, I recommend a literature about self- and cross-seeding on α-synuclein fibril (Watanabe-Nakayama, et al. ACS Nano 2020; 14: 9979-9989). It showed that cross-seeding modulated not only elongation rates but also the structures of the growing fibrils, and they suggested that α-synuclein sequence variants can produce different types of strains by self- or cross-seeding.

I also recommend literatures about α-synuclein oligomers. Ono K (Neurochem Res 2017; 42: 3362-3371) emphasized oligomer hypothesis in α-synucleinopathy. Ito, et al. (NPJ Parkinsons Dis 2023; 28: 139) revealed α-synuclein oligomers causes neuronal injury and death via cellular transmission and direct plasma membrane damage.

Author Response

Response to Reviewer 4 Comments

We are grateful to the Reviewer for the care with which our manuscript was read and for the constructive criticism and valuable suggestion. Taking into consideration the suggestions provided, we present an updated version of the article entitled Alpha-Synuclein Contribution to Neuronal and Glial Damage in Parkinson’s Disease. Essential changes were made to the text of our manuscript according to the comments and suggestions provided. In the revised version of the manuscript all changes made are highlighted in yellow. We confirm that all authors listed on the manuscript concur with the submission in its revised form. We hope that the revisions in the manuscript and our accompanying responses have made the manuscript suitable for publication in International Journal of Molecular Sciences.

Herein, we explain the revisions that were made based on the comments and recommendations.

General remark: This is a well-written review. However, I think it would be a more excellent paper if you add some references.

Response: We wish to thank the Reviewer for the positive feedback.

Point 1: Regarding α-synuclein aggregation, I recommend a literature about self- and cross-seeding on α-synuclein fibril (Watanabe-Nakayama, et al. ACS Nano 2020; 14: 9979-9989). It showed that cross-seeding modulated not only elongation rates but also the structures of the growing fibrils, and they suggested that α-synuclein sequence variants can produce different types of strains by self- or cross-seeding.

Response 1: We thank the Reviewer for their valuable suggestion. Thanks to the Reviewer’s comment, we have now added this information to the manuscript as follows:

(…) Furthermore, self-templating or cross-seeding of WT or mutant αSyn not only modulates the elongation rate, but also the structure of the growing fibrils, which gives rise to conformationally different strains [266]. (…)

Point 2: I also recommend literatures about α-synuclein oligomers. Ono K (Neurochem Res 2017; 42: 3362-3371) emphasized oligomer hypothesis in α-synucleinopathy. Ito, et al. (NPJ Parkinsons Dis 2023; 28: 139) revealed α-synuclein oligomers causes neuronal injury and death via cellular transmission and direct plasma membrane damage.

Response 2: We greatly appreciate the Reviewer’s recommendation. As suggested by the Reviewer, the indicated literature data have now been included in the manuscript in the following lines:

(…) Albeit small αSyn oligomers are considered critical species driving PD progression (as described in depth in [82]), their direct study poses many difficulties given their transient, metastable characteristics and high level of heterogeneity. (…)

(…) It has recently been proposed that high-molecular-weight αSyn oligomers exhibit high neurotoxicity by directly damaging plasma membrane integrity and inducing the extrinsic apoptotic pathway, which highlights their potential as a plausible target for the development of PD-modifying therapies [260].

Round 2

Reviewer 1 Report

Comments and Suggestions for Authors

No